Responses to simulated nitrogen deposition by the neotropical epiphytic orchid Laelia speciosa

Díaz-Álvarez Edison A. 1 2
Lindig-Cisneros Roberto 2
de la Barrera Erick 2 delabarrera@unam.mx
1 Posgrado en Ciencias Biológicas, Universidad Nacional Autónoma de México, Ciudad Universitaria , México, Distrito Federal , Mexico
2 Instituto de Investigaciones en Ecosistemas y Sustentabilidad, Universidad Nacional Autónoma de México , Morelia, Michoacán , Mexico
Sandhu Harpinder
Electronic publication date: 2015 Jun 23
Publication date: 2015
Volume: 3
Electronic Location ID: e1021
Received 2015 Feb 17; Accepted 2015 May 24
Copyright: © 2015 Díaz-Álvarez et al.
Copyright year: 2015
Copyright holder: Díaz-Álvarez et al.
License: This is an open access article distributed under the terms of the Creative Commons Attribution License, which permits unrestricted use, distribution, reproduction and adaptation in any medium and for any purpose provided that it is properly attributed. For attribution, the original author(s), title, publication source (PeerJ) and either DOI or URL of the article must be cited.
License URL: https://creativecommons.org/licenses/by/4.0/

Keywords: Acid rain, Biodiversity loss, CAM, Conservation physiology, δ15N, Nitrogen pollution, Stable isotopes, Global change, Plant nutrition

Funding: Dirección General del Personal Académico PAPIIT IN224910 RN204013 Instituto de Investigaciones en Ecosistemas y Sustentabilidad (IIES), UNAM Consejo Nacional de Ciencia y Tecnología, México We thank funding by Dirección General del Personal Académico (PAPIIT IN224910 and RN204013), UNAM and institutional funds from Instituto de Investigaciones en Ecosistemas y Sustentabilidad (IIES), UNAM. EADA was funded by a generous graduate research fellowship from Consejo Nacional de Ciencia y Tecnología, México. The funders had no role in study design, data collection and analysis, decision to publish, or preparation of the manuscript.

==============================
Potential ecophysiological responses to nitrogen deposition, which is considered to be one of the leading causes for global biodiversity loss, were studied for the endangered endemic Mexican epiphytic orchid, Laelia speciosa, via a shadehouse dose-response experiment (doses were 2.5, 5, 10, 20, 40, and 80 kg N ha−1 yr−1) in order to assess the potential risk facing this orchid given impending scenarios of nitrogen deposition. Lower doses of nitrogen of up to 20 kg N ha yr−1, the dose that led to optimal plant performance, acted as fertilizer. For instance, the production of leaves and pseudobulbs were respectively 35% and 36% greater for plants receiving 20 kg N ha yr−1 than under any other dose. Also, the chlorophyll content and quantum yield peaked at 0.66 ± 0.03 g m−2 and 0.85 ± 0.01, respectively, for plants growing under the optimum dose. In contrast, toxic effects were observed at the higher doses of 40 and 80 kg N ha yr−1. The δ13C for leaves averaged −14.7 ± 0.2‰ regardless of the nitrogen dose. In turn, δ15N decreased as the nitrogen dose increased from 0.9 ± 0.1‰ under 2.5 kg N ha−1yr−1 to −3.1 ± 0.2‰ under 80 kg N ha−1yr−1, indicating that orchids preferentially assimilate NH4+ rather than NO3− of the solution under higher doses of nitrogen. Laelia speciosa showed a clear response to inputs of nitrogen, thus, increasing rates of atmospheric nitrogen deposition can pose an important threat for this species.

Introduction

Anthropogenic atmospheric nitrogen deposition is considered among the leading global causes of biodiversity loss (Vitousek, 1994; Chapin et al., 2000; Sala et al., 2000). While nitrogen is an essential nutrient for all living organisms, its accelerated release to the atmosphere and ultimate deposition has caused saturation of various ecosystems around the world, leading to significant biodiversity loss by direct toxicity, acidification, and nutrient imbalances between nitrogen and other major nutrients (Aber et al., 1989; Bauer et al., 2004; Le Bauer & Treseder, 2008; Bobbink et al., 2010; Templer, Pinder & Goodale, 2012). Most studies regarding the effects of nitrogen deposition on biodiversity have been conducted in the USA and Europe, while studies from megadiverse countries are scant (Bobbink et al., 2010). Considering that the latter countries tend to have developing economies and accelerated industrialization processes, it is urgent to determine the effects that current and future nitrogen deposition rates may have on their local biodiversities (Austin et al., 2013).

A life-form particularly susceptible to the noxious effects of nitrogen deposition are epiphytic plants, such as certain species of orchids and bromeliads, given their reliance on atmospheric sources for nutrients and water (Zotz & Asshoff, 2010; Zotz et al., 2010; Mondragón, Valverde & Hernández-Apolinar, 2015). In this respect, Laelia speciosa (Kunth.) Shltr. (Orchidaceae) is an endemic, endangered orchid from central Mexico that has a cultural importance in Michoacán. Not only the plant is collected for its attractive flowers, but juice is extracted from its pseudobulbs and mixed with maize cane pith to produce a paste that is used for the production of sacred art in West Central Mexico (Soto-Arenas & Solano-Gómez, 2007). In addition to extractive pressure, this species faces environmental challenges considering that oak forests, to which this species is restricted, are likely to be severely reduced during the present century (Villers-Ruiz & Trejo-Vazquez, 2000; Rehfeldt et al., 2012). This study assessed whether nitrogen deposition can also pose a threat to this species. However, because current rates of nitrogen deposition are rather low within the area of distribution for L. speciosa (Díaz-Álvarez et al., 2014), it was deemed necessary to conduct a shadehouse dose-response experiment to determine the effects of potential future nitrogen deposition on this plant.

Indeed, the purpose of this study was to determine some ecophysiological responses of the endangered neotropical epiphytic orchid Laelia speciosa by means of a dose–response shadehouse experiment, in which, the organ production, chlorophyll content, chlorophyll fluorescence, carbon and nitrogen content and isotopic signatures were evaluated for assessing the potential risk that increasing rates of nitrogen deposition pose for this species.

Materials and Methods

Plant material

Laelia speciosa is a sympodial epiphytic orchid with big and showy flowers that have pink to lilac-purple petals and a white lip. Flowers are produced during the spring, while an annually produced carbon-storing pseudobulb develops during the summer. Laelia speciosa grows in sub-humid temperate climates of central Mexico, between 1250 and 2500 m where its predominant phorophyte Quercus deserticola is also found (Soto-Arenas & Solano-Gómez, 2007).

Two-year old plants of Laelia speciosa obtained by in vitro propagation were transferred into 2L plastic pots containing tezontle (particles were 2–5 cm in diameter), a very porous volcanic rock that is extensively utilized for gardening and hydroponic horticulture given its suitable physicochemical properties (Vargas-Tapia et al., 2008; Yañez-Ocampo et al., 2009). Organic matter was removed from this tezontle by submersion in a SO4H2 aqueous solution (50% v/v) followed by a double rinse with deionized, distilled water. The procedure was repeated thrice. The pots were placed in a shadehouse for 2 years at Universidad Nacional Autónoma de México, Campus Morelia (19°38′55.9″N; 101°13′45″W; 1967 m, mean annual temperature of 18.3 °C, annual precipitation 773 mm; Servicio Meteorológico Nacional, 2011), where they were watered every other week until the start of the experiment. A total of 120 plants were selected at random and assigned to one of six groups, each with 20 plants, which received different doses of nitrogen. At the start of the experiment, the plants had 4 pseudobulbs and one leaf (15 cm in length).

Nitrogen deposition scenarios

Starting on 1 October 2011, the plants were watered weekly over two months with 50 ml of a modified Hoagland No. 2 solution from which the nitrogen was omitted to be able to simultaneously supply suitable amounts of nutrients and manipulate the dose of nitrogen dispensed to plants (Hoagland & Arnon, 1950; Nobel & de la Barrera, 2002). At the end of this period, six simulated atmospheric deposition scenarios were applied by adding 1, 4, 8, 16 or 32 mM of NH4NO3 to the watering solution, equivalent to 2.5, 10, 20, 40, or 80 kg of N ha−1 yr−1 respectively. In this respect, a threshold for plant physiological damage has been observed at 20 kg N ha−1 yr−1, while rates of 40 kg N ha−1 yr−1 are common in certain parts of Mexico City (Britto & Kronzucker, 2002; Secretaria del Medio Ambiente del Gobierno del Distrito Federal, 2012). The range of doses considered was chosen to establish the threshold for physiological damage for L. speciosa, and to determine the effects of deposition rates that are likely to occur during the present century. All amounts were calculated according to the area of the pot of 201 cm2. Weekly applications of 50 ml of the experimental solutions were conducted over 26 weeks, from December 2011 to June 2012. This period corresponds to the growth season and reproductive development of Laelia speciosa (Halbinger & Soto, 1997; Soto-Arenas, 1994; Soto-Arenas & Solano-Gómez, 2007). Irrigation was carried out on the whole surface of the pot, the coarse substrate allowed the complete drainage of the nutrient solution, simulating what occurs in the canopy.

Physiological responses

Organ production

The emergence of flowers, which are displayed for a few weeks, was recorded weekly. In turn, the total production of new leaves and pseudobulbs, which are persistent, was recorded at the end of the experiment.

Chlorophyll content

Leaf discs were obtained with a cork borer (12-mm in diameter) from 5 plants per nitrogen deposition scenario to determine the concentration of chlorophyll a, chlorophyll b, and total chlorophyll in the plant tissue. The photosynthetic pigments were extracted by macerating leaf tissue with a chilled (3 °C) mortar and pestle in an aqueous solution of acetone (80% v/v) and brought to a final volume of 20 ml. Absorbance was measured at 663 and 646 nm with an EZ 301 spectrophotometer (Perkin Elmer, Waltham, Massachusetts, USA). Chlorophyll concentration was calculated following Lichtenthaler (1987).

Chlorophyll fluorescence (Fv/Fm)

The maximum yield of the photosystem II (the ratio of variable to maximum fluorescence; Fv/Fm) was measured with an Opti-Science 05-30p Fluorometer (Hudson, New Hampshire, USA). Measurements were carried out before dawn, a common practice in plant ecophysiology (Maxwell & Johnson, 2000), for the leaves of five individuals per dose of nitrogen on 29 June and 2 July 2012.

Carbon and nitrogen content and isotopic composition

The leaves of plants growing under different nitrogen doses were collected on 4 July 2012 and dried at 80 °C in a gravity convection oven until reaching constant weight. This temperature has been found to be adequate for tropical succulents, whose membrane proteins can withstand temperatures that are substantially higher than for non-succulent species without incurring in damage (Nobel & de la Barrera, 2002; Drennan, 2009). The dried leaves were ground to a fine powder in a ball mill (Retsch MM300; Retsch, Vienna, Austria), wrapped into tin capsules (Costech Analytical, Inc., Valencia, California, USA), and weighed with a microbalance (0.01 mg, Sartorius, Göttingen, Germany). For each sample, the carbon and nitrogen content, as well as their isotopic proportions, were determined at the Stable Isotope Facility, University of Wyoming (Laramie, Wyoming, USA), with a Carlo Erba EA 1110 elemental analyzer (Costech Analytical Inc., Valencia, California, USA) attached to a continuous flow isotope ratio mass spectrometer (Finnigan Delta Plus XP, Thermo Electron Corp, Waltham, Massachusetts). Carbon and nitrogen isotope ratios, reported in parts per thousand, were calculated relative to the Vienna Pee Dee Belemnite (V-PDB) or atmospheric air standards, respectively. The analytical precision for δ13C was ±0.03‰(SD) and ±0.06‰(SD) for δ15N. The natural abundances of 13C and 15N were calculated as: δ13C‰ versus V-PDB=Rsample/Rstandard−1×1000

δ15N‰ versus at-air=Rsample/Rstandard−1×1000

where, R is the ratio of 13C/12C for carbon and 15N/14N for nitrogen isotope abundance for a given sample (Ehleringer & Osmond, 1989; Evans et al., 1996).

Statistical analyses

The effect of the simulated nitrogen deposition on organ production for Laelia speciosa was evaluated by means of a Kruskal-Wallis non-parametric ANOVA, because normality of data was not satisfied, followed by post-hoc Tukey tests (P ≤ 0.05). In turn, differences in the response of chlorophyll content, chlorophyll fluorescence, carbon and nitrogen content, and δ13C and δ15N, which achieved normality, were evaluated with a one-way ANOVA followed by the Holm-Sidak post-hoc test (P ≤ 0.05). All analyses were conducted on SigmaPlot 12 (Systat Software Inc., San Jose, California, USA).

Results

Organ production

After 26 weeks of watering the plants with different doses of nitrogen, the production of new organs was greater for those individuals that received 20 kg N ha−1 yr−1 than for those individuals receiving other nitrogen doses (Table 1; Fig. 1). In particular, 1.0 ± 0.1 leaves were produced per plant over the course of the experiment under most doses, except for the plants that received 20 kg N ha−1 yr−1, which produced 35% more leaves (P ≤ 0.001). Similar was the case for the 0.9 ± 0.1 pseudobulbs produced per plant under most doses, except for the plants that received 20 kg N ha−1 yr−1, which produced 36% more pseudobulbs (P ≤ 0.001). In contrast, flowering was not significantly influenced by nitrogen dose (P = 0.077), with a production of 0.3 ± 0.04 flowers per plant over the course of the experiment (Table 1; Fig. 1).

Figure 1 Organ production.

Number of new leaves (open bars), pseudobulbs (right hatched bars) and flowers (left hatched bars) that developed on plants of Laelia speciosa that were watered with different doses of nitrogen. Data are shown as mean ± S.E (n = 20 plants per dose of nitrogen). Different letters indicate significant differences (p < 0.05) for organs.

Table 1 Statistical analyses.

Kruskal-Wallis one-way ANOVA and parametric one-way ANOVA for the responses of Laelia speciosa individuals growing in a shadehouse under various rates of simulated nitrogen deposition.

	Response to nitrogen dose	
	d.f.	F	P	
Leaves	5	8.47	0.001	
Pseudobulbs	5	7.04	0.001	
Flowers	5	1.94	0.077	
Total chlorophyll	5	15.68	0.001	
Chla	5	6.67	0.001	
Chlb	5	10.47	0.001	
Fv/Fm	5	82.5	0.001	
Carbon content	5	6.44	0.001	
Nitrogen content	5	177.5	0.001	
δ15N	5	15.68	0.001	
δ13C	5	2.65	0.057	

Chlorophyll fluorescence

The quantum efficiency of photosystem II (Fv/Fm) was similar among the groups of orchids that received up to 20 kg N ha−1 yr−1 amounting to 0.8, while a significant decrease of 23% was observed for plants irrigated with higher concentrations of nitrogen (P ≤ 0.001; Table 1; Fig. 2A).

Figure 2 Ecophysiological responses for leaves of L. speciosa to simulated nitrogen deposition.

(A) Ratio of variable to maximum chlorophyll fluorescence; (B) Tissue content (area basis) for total chlorophyll (circles), chlorophyll-a (triangles), and chlorophyll-b (square); (C) Carbon and (D) nitrogen content (dry mass basis) and (E) δ15N. Data are shown as mean ± S.E. (n = 5 plants per dose of nitrogen). For each panel, different letters indicate significant differences (P < 0.05).

Chlorophyll content

Total chlorophyll content for the leaf tissue of Laelia speciosa increased as the nitrogen dose increased, peaking at 0.7 ± 0.0 g m−2 for plants irrigated with 20 kg N ha−1 yr−1 (P ≤ 0.001; Table 1; Fig. 2B), while the higher doses of nitrogen resulted in a 38% reduction of the pigment. Similarly, the chlorophyll a concentration of 0.5 ± 0.4 g m−2 was the maximum for plants growing under 20 kg N ha−1yr−1, and it was 30% lower under all other nitrogen doses (P ≤ 0.001). In turn, chlorophyll b did not respond to nitrogen, averaging 0.1 ± 0.0 g m−2 regardless of the dose under which plants grew (Table 1; Fig. 2B).

Carbon and nitrogen content and isotopic composition

The carbon content of Laelia speciosa increased with the nitrogen dose peaking at 46.1 ± 0.3% (dry mass basis) at 20 and 40 kg N ha−1 yr−1 and decreased to 45.2 ± 0.3% at 80 kg N ha−1 yr−1 (P ≤ 0.001; Table 1; Fig. 2C).

The nitrogen content for Laelia speciosa also increased with the nitrogen dose. For the plants that received up to 10 kg N ha−1 yr−1 the nitrogen content averaged 1.2 ± 0.0% (dry mass basis), reaching 2.4 ± 0.0% at 80 kg N ha−1 yr−1 (P ≤ 0.001; Table 1; Fig. 2D).

The δ13C for leaves of Laelia speciosa averaged −14.7 ± 0.2‰ and did not change with the nitrogen dose (P = 0.057; Table 1). In contrast, the leaf δ15N significantly decreased at higher nitrogen doses. The δ15N averaged 0.9 ± 0.1‰ for plants that received up to 10 kg N ha−1 yr−1, a δ15N similar to the δ15N of 1.1 ± 0.1‰ measured for the NH4NO3 utilized for the nutrient solution. The higher doses of nitrogen led to significant decreases of δ15N, reaching the minimum of −3.1 ± 0.2‰ for plants growing under 80 kg N ha−1 yr−1 (P ≤ 0.001; Table 1; Fig. 2E).

Discussion

An intermediate nitrogen dose of 20 kg N ha−1 yr−1 was the most favorable for the production of new organs by Laelia speciosa. Lower doses did not improve plant development substantially but higher doses were inhibiting. In this respect, while nitrogen availability may increase leaf production and growth, large quantities of nitrogen limit the availability of other nutrients, restricting the plant’s ability to produce foliar mass (Evans, 1989; Asner, Seastedt & Townsend, 1997; Aber et al., 1998; Sánchez et al., 2000; Zotz & Asshoff, 2010; Díaz-Álvarez et al., 2014). Such behavior was observed for Laelia speciosa that showed a substantial reduction in the production of new organs, suggesting noxious effects of the simulated nitrogen deposition. The effect of nitrogen fertilization on Cymbidium hybrids is an increased pseudobulb production (Barman et al., 2004). In turn, pseudobulb growth for Dendrobium nobile peaks at nitrogen doses of 1.9 mM (Bichsel, Starman & Yin-Tung, 2008).

Total chlorophyll content is proportional to the content of nitrogen in leaves, which typically ranges between 0.4 and 0.5 g m−2 (Evans, 1989; Nobel, 1999; Nobel & de la Barrera, 2002). Indeed, for Laelia speciosa, chlorophyll content increased with the dose of nitrogen, suggesting that this plant was able to assimilate and utilize the supplied nitrogen for the production of photosynthetic pigments. However, the higher doses also resulted in a drastic decrease of the chlorophyll content, as has been documented for other plant species (Baxter, Emes & Lee, 1992; Majerowicz et al., 2000; Lin et al., 2007; Arróniz-Crespo et al., 2008; Ying-Chun et al., 2010). Such a decrease in the chlorophyll content can be explained by the resulting imbalance of the nitrogen to magnesium ratio in the leaf (Nakaji et al., 2001; Wortman et al., 2012). Excessive nitrogen in the cell promotes release of protons (H+) and accumulation of phenolic compounds and hydrogen peroxide, as a result, the pH can be altered impeding chlorophyll production and loss of Mg2+ (Mangosá & Berger, 1997; Sánchez et al., 2000; Britto & Kronzucker, 2002). Changes in chlorophyll content for Laelia speciosa were accompanied by changes in the efficiency of photosystem II, which can be attributed to oxidative stress in the thylakoids that results in the blockage of electron transport to the oxidation site, as a consequence of low available energy for photosynthesis (Maxwell & Johnson, 2000; Poorter, 2000; Hogewoning & Harbinson, 2007; Lichtenthaler et al., 2007; Baker, 2008; Calatayud et al., 2008; Guidi & Degl’Innocenti, 2008; Massacci et al., 2008).

Plants tend to increase their rates of carbon fixation when nitrogen is added (Brown et al., 1996; Bauer et al., 2004; Le Bauer & Treseder, 2008). However, under conditions of chronic nitrogen additions the photosynthetic capacity is inhibited because most of the excess nitrogen is not invested into the primary processes of carboxylation (Brown et al., 1996; Bauer et al., 2004). This also causes an increase and later reduction in carbon content for plants subjected to increasing doses of nitrogen, as was observed here for Laelia speciosa. However, the observed δ13C values for Laelia speciosa which were within the range for CAM plants, did not change under the different nitrogen doses utilized, contrasting with δ13C measured for C3 plants subjected to supplementary nitrogen that became increasingly negative (Raven & Farquhar, 1990; Magalhaes, Huber & Tsai, 1992).

Isotopic discrimination against 15N increases in plants as the nitrogen availability increases because its assimilation is more energetically costly than for the more abundant 14N. This so called isotopic effect results in δ15N values of the product that are lower than those of the substrate (Evans, 2001; Kolb & Evans, 2003; Ariz et al., 2011). The observed discrimination against 15N for Laelia speciosa leaves has also been observed for various species, such as Oryza sativa, Pinus sylvestris, and Trapa japonica, species which discriminate between 0.9 and 13‰ when supplied with increasing doses of nitrogen in form of NH4+ (Yoneyama et al., 1991; Högberg et al., 1999; Yoneyama et al., 2001; Maniruzzaman & Asaeda, 2012). When the nitrogen source is NH4+, this compound is directly assimilated by the plant cell as amino acids and the involved enzyme, glutamine-synthetase, can discriminate up to 17‰. On the contrary, plants watered with NO3− have positive δ15N values that have been associated with nitrogen lost via root efflux and exudates or loss of NH3 through the stomata, processes that favor the lighter isotope (O’Deen, 1989; Yoneyama et al., 2001; Ariz et al., 2011).

Laelia speciosa showed a clear response to increasing doses of nitrogen. Doses of up to 20 kg N ha−1 year−1 enhanced its physiological performance, while higher doses were toxic. The rates of nitrogen deposition in México, where Laelia speciosa is endemic, could exceed 25 kg N ha−1 year−1 by mid-century (Galloway et al., 2004; Phoenix et al., 2006; Galloway et al., 2008). As a result, nitrogen deposition poses an actual threat for the persistence of this endangered species as other components of global change represent for many other epiphytic vascular plants (Zotz et al., 2010; Mondragón, Valverde & Hernández-Apolinar, 2015). Future works should consider the effects of nitrogen deposition on wild populations of this and other tropical epiphytic plants. A better understanding of the effects of increasing nitrogen deposition from human activities is of urgent importance, as species ecophysiological response, as those studied here, may be affected, with potentially negative consequences in ecosystem biodiversity and function.

Supplemental Information

Supplemental Information 1 Statistical analysis

Click here for additional data file.

We thank Dr. I Ávila Díaz for providing the plants utilized in this study; the personnel of the IIES Botanical Gardens for maintenance of the plants before the experiment, especially Ms. MD Lugo and Mr. J. Martinez Cruz; and Dr. DG Williams at the University of Wyoming for his guidance during the elemental and isotopic analyses of the samples.

Additional Information and Declarations

Competing Interests

Author Contributions

The authors declare there are no competing interests.

Edison A. Díaz-Álvarez conceived and designed the experiments, performed the experiments, analyzed the data, wrote the paper, prepared figures and/or tables, reviewed drafts of the paper.

Roberto Lindig-Cisneros conceived and designed the experiments, contributed reagents/materials/analysis tools, reviewed drafts of the paper, supervised plant nutrition experiments.

Erick de la Barrera conceived and designed the experiments, contributed reagents/materials/analysis tools, wrote the paper, prepared figures and/or tables, reviewed drafts of the paper.

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
