# Peer review of "Responses to simulated nitrogen deposition by the neotropical epiphytic orchid Laelia speciosa"

_PeerJ, doi:10.7717/peerj.1021_

## Round 0.1 · original submission · Major Revisions

· Academic Editor

Major Revisions

As pointed by reviewers, and in my opinion, it is an interesting study that warrants publication in a scientific journal. Moreover, the manuscript has improved considerably from the first set of reviews. However, as noticed in this second set of review, there are still some sections (particularly introduction and discussion) and several comments raised by both reviewers, that needs to be addressed before this manuscript could be acceptable for publication.

Please pay careful attention to the suggestions by both reviewers in your revisions. I look forward to receiving a revised version.

Reviewer 1 ·

Basic reporting

The manuscript addresses the influence of the increment on N deposition over an orchid species. The authors followed the reviewers´suggestions and made a series changes that turned the manuscript, in a general way, much more interesting to the reader. Some points need to be reconsidered, though. First of all, English needs to be revised again by a native-speaker. The authors should also standardize the use of 'nitrogen'/'carbon' or 'N'/'C' throughout the manuscript.

Once again, I will talk about the Abstract and Introduction sections here, since there are no available boxes for them below.

The description of the results is confusing at both abstract and Results section. Still regarding the abstract, the authors should present the dose used during the greenhouse glass experiment. Once again, the Introduction section needs to follow a more logical sequence. The new version is poorly detailed at some points; unfortunately, I missed more information about the usefulness to do a greenhouse glass experiment or to use an endangered orchid species as plant model.

Experimental design

The Methods section is better fitted now, with a good description about the methodology used. The removal of the section comparing the orchids from the two sites made the manuscript more clear. However, I do not understand why the authors evaluate variation on organ production (flowers, pseudobulbs and leaves) with increasing N inputs if they do not discuss this issue at the end of the manuscript. Thus, they should either exclude this subsection at the Methods and Results section or mention it during the Discussion section.

Still regarding the Methods, I keep not thinking that drying leaves at 80oC is proper when determining elemental and isotopic compositions of a plant material. Even such leaves are succulent, as the authors correctly pointed. So, my concern keeps as a suggestion for their next studies.

In relation to the statistical procedures, I wonder why the authors evaluated data distribution, found out that data normality was ok and still decided to use a non-parametric test. If they decided to use this kind of test instead of a common ANOVA, it would be preferable if they explain their reasons.

Validity of the findings

The Results section should be written in a simpler way. The way the authors presented, it is a little bit confusing (for example, lines 160-166). After all findings, the authors should present the P-value of the analysis, not only in table of figures. One last comment about the results: I believe that the removal of the lines from the figures would ease the visualization of the mean values and their variation.

The Discussion section, as I commented earlier, lacks any topic about the growth/production of new organs. Once again, they should either exclude this topic at the Methods and Results section or mention it during the Discussion section.
The part discussing the preferential use of NH4+ or NO3- should be improved. As increased N-availability may lead to higher residual δ15N values in a system, the authors should discuss about the fractionation processes during N transformation in soils, and also present the isotopic composition of the N coumponds in soils. They should make a deeper discussion about this preferential use of NH4+ they found.

Additional comments

No comments.

Reviewer 2 ·

Basic reporting

The authors present a very interesting data to answer an important question: what would be the effect of increased nitrogen deposition on the ecophysiology of a vascular epiphyte? However, I believe that the authors should make certain amendments before their item can be published.

First of all, despite that epiphytes are considered as one of groups most susceptible to changes in atmospheric deposition, the authors make no reference to the subject, and didn´t compare their results with one of the few case studies inside of epiphytes (see list of suggested reading). I believe that it would be very important to introduce this subject, both in the introduction and in the discussion section.

Furthermore it intrigues me that the author does not describe the species in the Materials and methods. Also in this section, it is important that the authors justify the doses of N for their treatments.

Experimental design

In the statistical part the author has to justify why use a parametric post-hoc test in a nonparametric test; and why used the Holm-Sidak post-hoc test instead of Tukey test.

Validity of the findings

In the results section is important to note that there are statistically significant differences in all measured parameters, except for the production of flowers and the C15. It is also very important that the authors note, that once the ANOVA given that there weren´t significant differences, not posteriori tests are performed.

Additional comments

suggested reading
Population ecology of epiphytic angiosperms: a review
D Mondragon, T Valverde, M Hernandez-Apolinar - Trop. Ecol, 2015
Growth of epiphytic bromeliads in a changing world: the effects of CO 2, water and nutrient supply G Zotz, W Bogusch, P Hietz, N Ketteler - Acta Oecologica, 2010 - Elsevier
Growth in epiphytic bromeliads: response to the relative supply of phosphorus and nitrogen G Zotz, R Asshoff - Plant Biology, 2010

Annotated reviews are not available for download in order to protect the identity of reviewers who chose to remain anonymous.
External reviews were received for this submission. These reviews were used by the Editor when they made their decision, and can be downloaded below.

---

## Round 0.2 · accepted · Accept

· Academic Editor

Accept

I am satisfied with the revised version as it has addressed the issues raised by both reviewers. It also reads well.